# A Simple Flow Injection Analysis–Tandem Mass Spectrometry Method to Reduce False Positives of C5-Acylcarnitines Due to Pivaloylcarnitine Using Reference Ions [note 1]

**DOI:** 10.3390/children9050694

**Published:** 2022-05-10

**Authors:** Takanari Hattori, Yoshitomo Notsu, Misa Tanaka, Miki Matsui, Tetsuo Iida, Jun Watanabe, Yoshimitsu Osawa, Seiji Yamaguchi, Shozo Yano, Takeshi Taketani, Hironori Kobayashi

**Affiliations:** 1Shimadzu Corporation, Kyoto 604-8511, Japan; t-hat@shimadzu.co.jp (T.H.); t-iida@shimadzu.co.jp (T.I.); jun_wtnb@shimadzu.co.jp (J.W.); 2Laboratories Division, Shimane University Hospital, Izumo 693-8501, Japan; ynotu34@med.shimane-u.ac.jp (Y.N.); syano@med.shimane-u.ac.jp (S.Y.); 3Department of Pediatrics, Faculty of Medicine, Shimane University, Izumo 693-8501, Japan; mstanaka@med.shimane-u.ac.jp (M.T.); matsui.miki027@med.shimane-u.ac.jp (M.M.); y-osawa@gunma-u.ac.jp (Y.O.); seijiyam@med.shimane-u.ac.jp (S.Y.); ttaketani@med.shimane-u.ac.jp (T.T.)

**Keywords:** isovaleric acidemia, newborn screening, false positive, C5-acylcarnitine, isovalerylcarnitine, pivaloylcarnitine, mass spectrometry

## Abstract

Flow injection analysis–tandem mass spectrometry (FIA-TMS) has been applied in a first-tier test of newborn screening (NBS). Although isovalerylcarnitine (i-C5), which is a diagnostic indicator of isovaleric acidemia (IVA), is isobaric with pivaloylcarnitine (p-C5), 2-methylbutyrylcarnitine, and n-valerylcarnitine, these isomers cannot be distinguished by the FIA-TMS. There are many reports of false positives derived from p-C5 due to the use of pivalate-conjugated antibiotics. In this study, we developed a new FIA-TMS method to distinguish i-C5 and p-C5. We found that the intensity ratio of product ions for i-C5 and p-C5 was different in a certain range even under the same analytical conditions. The product ions with the most distinct differences in ionic intensity between the isomers and the collision energies that produce them were determined to be *m/z* 246.2 > 187.1 and −15 V, respectively. In addition to the quantification ion, a reference ion was defined, and the similarity of the i-C5 and p-C5 reference ion ratios (i-C5 score and p-C5 score, respectively) were used to estimate which isomer (i-C5 and p-C5) was responsible for elevated C5 acylcarnitine in dried blood spots (DBSs). As a result of analyses of 11 DBS samples derived from pivalate-conjugated antibiotics and four DBS samples from IVA patients using our method, it was found that our method was able to correctly determine the type of C5-acylcarnitine (i-C5 or p-C5) in the DBS samples. Implementation of this new FIA-TMS method into the current NBS protocol will allow for a reduction in false positives in IVA.

## 1. Introduction

Newborn screening (NBS) is an important public health program for improving children’s health in most developed and developing countries. This protocol has provided early diagnosis and intervention to inherited metabolic diseases, preventing severe health problems, including death. Amino acid and acylcarnitine analysis in dried blood spots (DBSs) using flow injection analysis–tandem mass spectrometry (FIA-TMS) is widely used in NBS. In FIA-TMS, a sample is injected into a mass spectrometer by flow injection without column separation, and amino acids and acylcarnitines are detected in multiple reaction monitoring (MRM) mode in a triple quadrupole mass spectrometer. FIA-TMS is suitable for a first-tier test because the time required for analysis is short (1–2 min), and many samples can be rapidly analyzed [1,2,3,4,5,6].

Isovaleric acidemia (IVA) is the first organic acidemia to be described by Tanaka et al. [7] and is one of the target diseases in NBS. IVA is caused by a deficiency in isovaleryl-CoA dehydrogenase, resulting in the accumulation of derivatives of isovaleryl-CoA, isovalerylcarnitine (i-C5), isovalerylglycine, and occasionally 3-hydroxyisovaleric acid. The clinical features of IVA include peculiar body smell and episodic attacks of vomiting, lethargy, metabolic acidosis, and even coma [8].

In the first-tier test using FIA-TMS, C5 is used as a marker metabolite of IVA [9,10,11]. i-C5 is isobaric with pivaloylcarnitine (p-C5), 2-methylbutyrylcarnitine (m-C5), and n-valerylcarnitine (v-C5). All of these isomers are identified as the same ‘C5′ in FIA-TMS. These isomers cannot be distinguished by FIA-TMS. Although m-C5 is a marker of short/branched chain acyl-CoA dehydrogenase deficiency (SBCADD), SBCADD is not included in NBS panels because it is an asymptomatic condition with unknown clinical consequences in the longer term [12,13]. v-C5 accumulates due to metabolic disorders involving fatty acids with odd-numbered carbon chains [14]. p-C5 can be present in DBSs due to newborn or maternal use of pivalate-containing antibiotics or maternal use of pivalate derivatives used as emollients in some nipple creams. Pivalate conjugates to free carnitine and becomes p-C5 [15,16,17,18,19,20]. There are many reports of false positives of elevated C5, which often are derived from p-C5. Unfortunately, the positive predictive value of IVA is considerably lower than that of other target diseases in NBS due to p-C5, leading to unnecessary recall and detailed examination. False positives can cause a nonoptimal allocation of healthcare resources and parental anxiety [21,22,23,24]. The positive rate of screening for IVA in our program was 0.001%, and all presumptive positive subjects were false positives. In the English Newborn Screening for isovaleric acidemia, Kohling et al. reported a positivity rate of 0.004%; 18 subjects were presumptive positive and 14 were false positive due to p-C5 [25]. Therefore, it is very important to distinguish p-C5 and other isomers of C5-acylcarnitine to reduce the false positives of IVA. To distinguish these isomers, liquid chromatography–tandem mass spectrometry (LC–MS/MS) with column separation is effective. In LC–MS/MS, isomers of C5-acylcarnitine are separated in a column, and each is detected by a triple quadrupole mass spectrometer. Some methods using LC–MS/MS with column separation have been reported as second-tier tests, but a simpler method was desired to avoid additional LC–MS/MS analysis.

In this study, we report a new FIA-TMS method for a first-tier test to distinguish i-C5 and p-C5 in DBSs with raised levels of C5-acylcarnitine. To date, few studies have been reported to distinguish the isomers of C5-acylcarnitine in a first-tier test using FIA-TMS. Therefore, we investigated methods to distinguish the isomers of C5-acylcarnitine, i-C5, and p-C5 using the specific product ion or the reference ion ratio under conventional FIA-TMS conditions. To confirm the effectiveness of this new FIA-TMS method in a first-tier test, DBS samples with high levels of C5-acylcarnitine were analyzed.

## 2. Materials and Methods

### 2.1. Design and Participants

This study was performed according to the guidelines of the Declaration of Helsinki and approved under study number 20211026-1 by the Institutional Review Committee of Shimane University Faculty of Medicine.

### 2.2. Regents

Isovaleryl-l-carnitine and 2-methylbutyryl-l-carnitine were purchased from Merck (Darmstadt, Germany). l-Pivaloylcarnitine was purchased from AptoChem (Montreal, QC, Canada). Valeryl-l-carnitine was purchased from Cambridge Isotope Laboratories, Inc. (Tewksbury, MA, USA). LC–MS-grade methanol, ultrapure water, ammonium formate, formic acid, and acetonitrile were purchased from FUJIFILM Wako Pure Chemical Corporation (Osaka, Japan). The mobile phase for FIA-TMS was provided using a NeoBase Kit (Perkin Elmer, Waltham, MA, USA).

### 2.3. Sample Preparation Procedure

DBS samples were prepared in accordance with standardized protocols of the nonderivatized method using a NeoBase kit. Briefly, a single 3.2 mm DBS punch disc was placed in each well of a 96-well assay plate. A total of 100 μL of the extraction solution containing an internal standard of acylcarnitines and amino acids was added to each well. The plate was shaken at 700 rpm at 45 °C for 45 min, and the supernatant was transferred to another plate for analyses.

### 2.4. Mass Spectrometric Analysis

A Nexera™ MP system coupled with an LCMS™-8050 triple quadrupole mass spectrometer (Shimadzu Corporation, Kyoto, Japan) was used for FIA-TMS. The mobile phase was 0.03% oxalic acid in water and methanol (22:78, *v*/*v*). The pump was operated as follows: 0.1 mL/min (0 min), 0.05 mL/min (0.1 min), 0.1 mL/min (0.65 min), and 0.5 mL/min (0.66–1 min). The injection volume of the samples was set at 1 µL. The ESI positive mode was used with an interface voltage of +4 kV. The temperatures of the desolvation line and heat block were set at 250 and 400 °C, respectively. The flow rates of the nebulizer gas and drying gas were set at 3 and 10 L/min, respectively. In the experiment detailed in Section 3.1, MS/MS data were acquired using the product ion scan mode for the *m/z* 246.2 precursor ion over the range of *m/z* 10–250. The collision energy (CE) was set at −10, −30, or −50 V. Standard solutions of i-C5, p-C5, m-C5, and v-C5 (100 μg/L) were used. In the experiments outlined in Section 3.2 and Section 3.4, MRM mode was used. The quantitative ion was *m/z* 246.2 > 85.0 (CE: −23 V), and the reference ion was *m/z* 246.2 > 187.1 (CE: −15 V). Standard solutions (10 μg/L) of i-C5, p-C5, m-C5, and v-C5 were used.

A Nexera™ X2 system coupled with an LCMS™-8030+ triple quadrupole mass spectrometer (Shimadzu Corporation, Kyoto, Japan) was used for LC–MS/MS. A CHIRALPAK ZWIX (+) column (250 mm × 4 mm, 3 µm, Daicel Corporation, Osaka, Japan) was used. The mobile phase was 25 mmol/L ammonium formate and 25 mmol/L formic acid in water and methanol (2:98, *v*/*v*). The pump was operated at 0.4 mL/min with isocratic elution. The injection volume of the samples was set at 5 µL. Positive electrospray ionization (ESI) was used with an interface voltage of +4.5 kV. The temperatures of the desolvation line and heat block were set at 250 and 400 °C, respectively. The flow rates of the nebulizer gas and drying gas were set at 3 and 15 L/min, respectively. MRM mode was conducted at *m/z* 246.2 > 85.0 (CE: −25 V).

A Nexera™ X3 system coupled with an LCMS™-9030 quadrupole time-of-flight mass spectrometer (Shimadzu Corporation, Kyoto, Japan) was used for LC-QTOFMS. A Shim-pack™ GIST C18 column (50 mm × 2.1 mm, 3 µm, Shimadzu Corporation, Kyoto, Japan) was used. The mobile phase was 0.1% formic acid in water and acetonitrile (50:50, *v*/*v*). The pump was operated at 0.2 mL/min with isocratic elution. The injection volume of the samples was set at 1 µL. Positive ESI was used with an interface voltage of +4 kV. The temperatures of the interface, desolvation line, and heat block were set at 300, 250, and 400 °C, respectively. The flow rates of the nebulizer gas, heating gas, and drying gas were set at 3, 10, and 10 L/min, respectively. The MS/MS data were acquired using MS/MS mode for the *m/z* 246.17 precursor ion over the range of *m/z* 10–300. The CE was set at 20 V. Standard solutions (10 mg/L) of i-C5 and p-C5 were used. Before analyses, external mass calibration using clusters of NaI was performed. ACD/MS Fragmenter (Advanced Chemical Development, Toronto, ON, Canada) was used to assign product ions of i-C5 and p-C5.

## 3. Results

### 3.1. Specific Product Ion of C5-Acylcarnitine

Although isomers of C5-acylcarnitine (i-C5, p-C5, m-C5, and v-C5) had different terminal structures in the acyl chain (Figure 1), these isomers could not be distinguished by the conventional FIA-TMS method. Therefore, we screened specific product ions that were generated from the acyl chain to distinguish i-C5 from other isomers. Figure 2 shows the product ion spectra of these isomers when the CE was set at −10, −30, or −50 V. Typical product ions of *m/z* 187.1, 85.0, 57.1, 41.1, and 29.1 were detected for all isomers. The specific product ion of these isomers of C5-acylcarnitine was not generated. This revealed that these isomers cannot be distinguished by the specific product ion.

### 3.2. Reference Ion Ratio

In MRM mode, the reference ion is used for identifying the molecule in addition to the quantitative ion for quantification. We found that the intensity of the reference ion relative to the quantitative ion was different for each isomer. Therefore, we investigated the reference ion ratios (the ratio of the peak intensity of the reference ion to the quantitative ion) for each isomer in various CEs. As shown in Figure 2, in order of peak intensity, *m*/*z* 246.2 > 85.0 (CE: −23 V) and *m*/*z* 246.2 > 187.1 (CE: −15 V) were determined as the quantitative ion and reference ion, respectively. The other ions (*m/z* 57.1, 41.1, 29.1) were not appropriate as product ions in terms of peak intensity and reproducibility. MRM chromatograms of i-C5, p-C5, m-C5, and v-C5 are shown in Figure 3. The results of continuous analyses (n = 6) revealed that the reference ion ratios of i-C5, p-C5, m-C5, and v-C5 were 19.97 ± 0.61%, 27.25 ± 0.56%, 22.89 ± 0.43%, and 18.07 ± 0.77%, respectively. There was no significant difference in the reference ion ratios for i-C5, m-C5, and v-C5, but the reference ion ratio for p-C5 was higher than that of the other isomers, suggesting that p-C5 could be distinguished from the other isomers by its reference ion ratio. Normally, most of the increased C5-acylcarnitine in NBS is due to i-C5 and sometimes to p-C5. It is very rare that the differentiation of m-C5 or v-C5 becomes a problem [25], and their identification is not important for the initial test. Therefore, we examined whether the similarity of the reference ion ratios of i-C5 and p-C5 could be used as an index to estimate which of the two was dominant in C5 in the sample. Similarities of the reference ion ratio of i-C5 and p-C5 were described as the i-C5 score and p-C5 score by the following equation:i-C5 score = 100 − |R_DBS_ − R_i-C5_|/R_i-C5_ × 100
p-C5 score = 100 − |R_DBS_ − R_p-C5_|/R_p-C5_ × 100
where R_DBS_ is the reference ion ratio when the DBS sample is analyzed, R_i-C5_ is the reference ion ratio when the standard solution of i-C5 is analyzed, and R_p-C5_ is the reference ion ratio when the standard solution of p-C5 is analyzed.

### 3.3. Structural Analysis of Product Ions Using LC–QTOFMS

To investigate the cause of the difference in the reference ion ratio between i-C5 and p-C5, structural analysis of the product ions was performed by liquid chromatography–quadrupole time-of-flight tandem mass spectrometry (LC–QTOFMS). Figure 4 shows the product ion spectra of i-C5 and p-C5. *m/z* values of 187.0963, 144.1017, 85.0275, and 57.0331 were detected for product ions. The structure of these product ions was estimated from the structure of i-C5 and p-C5 and the accurate mass of these product ions. From the structure of these product ions and Figure 2, it was found that more *m/z* 85.0275 was generated from *m/z* 246.1698 and 187.0963 with increasing CE.

### 3.4. Identification of i-C5 and p-C5 in DBSs

To investigate the efficacy of the i-C5 score and p-C5 score in distinguishing i-C5 and p-C5 in DBSs, 11 DBS samples (cases 1–11) with raised levels of C5-acylcarnitine derived from pivalate-containing antibiotics and 4 DBS samples (cases 12–15) with high levels of i-C5 derived from IVA patients were analyzed by LC–MS/MS (n = 4) and FIA-TMS (n = 4). LC–MS/MS analysis was performed to determine the proportions of i-C5, p-C5, m-C5, and v-C5 in these DBS samples. The proportion of i-C5, p-C5, m-C5, and v-C5; the concentration of C5-acylcarnitine by FIA-TMS analysis; the i-C5 score; and the p-C5 score are shown in Table 1. In all 11 DBS samples derived from cases who had pivalate-containing antibiotics, the p-C5 score was higher than the i-C5 score. Analyses of four DBS samples derived from IVA patients showed higher i-C5 scores than p-C5 scores. As a result of Mann–Whitney U test, the p-value of both i-C5 score and p-C5 score in cases 1–11 and 12–15 was 0.00496. These results indicated that our method was able to correctly determine the type of C5-acylcarnitine in the DBS samples. The correlation coefficients between the percentage of i-C5 in DBS and i-C5 score and between the percentage of p-C5 in DBS and p-C5 score were 0.93 and 0.85, respectively, indicating a strong correlation.

The accuracy of the i-C5 score and p-C5 score was investigated. The number of injections per sample required to measure the i-C5 score and p-C5 score with an accuracy of ± 5 (confidence interval, 95%) was calculated by EZR [26]. For the calculation, standard deviations obtained by continuous analyses (n = 30) of the mixture of 11 DBS samples that were derived from pivalate-containing antibiotics and the mixture of 4 DBS samples that were derived from IVA patients were used. In analyses of the mixture of 11 DBS samples, the standard deviations of the i-C5 score and p-C5 score were 3.31 and 1.87, respectively. The numbers of injections per sample required to measure the i-C5 score and p-C5 score with an accuracy of ± 5 (confidence interval, 95%) were 2 and 1, respectively. In analyses of the mixture of four DBS samples, the standard deviations of the i-C5 score and p-C5 score were 1.75 and 1.30, respectively. The numbers of injections per sample required to measure the i-C5 score and p-C5 score with an accuracy of ±5 (confidence interval, 95%) were both 1. Therefore, it was suggested that this new FIA-TMS method is effective in distinguishing positive or false-positive IVA in the primary test because the i-C5 score and p-C5 score can be obtained with high accuracy by conducting two analyses per sample, which corresponds to the initial test and confirmation test in a routine test.

## 4. Discussion

In the present study, we developed a new FIA-TMS method to reduce false positives of IVA due to p-C5 and demonstrated that the method could be applied to routine testing of NBS. To discriminate the isomers of C5 acylcarnitine, we first investigated whether there were specific product ions for each isomer, but we could not find any specific product ions that distinguished the four isomers, i-C5, P-C5, m-C5, and v-C5. However, it was found that the amounts of product ions for i-C5 and p-C5, the most important isomers in routine NBS, differed with a certain trend. The product ions with the most distinct differences in ionic intensity between the isomers and the collision energies that produce them were determined to be *m*/*z* 246.2 > 187.1 and −15 V, respectively. In conventional acylcarnitine analysis by FIA-TMS, the product ion with the highest ionic intensity is used for quantification (*m*/*z* 246.2 > 85.0). In our method, in addition to the routine quantification, a reference ion was defined, and the similarity of the i-C5 and p-C5 reference ion ratios (i-C5 score and p-C5 score, respectively) was used to estimate which isomer (especially i-C5 and p-C5) is responsible for C5 acylcarnitine in DBSs. In the identification using the reference ion ratio, it was found that the reference ion ratio of p-C5 was higher than that of other isomers of C5-acylcarnitine. It was suggested that the difference in the reference ion ratio was due to the terminal structure in the acyl chain and the CE required to decompose to *m*/*z* 85. p-C5 is less likely to be decomposed than the other isomers because p-C5 has a bulky three-dimensional structure in which three methyl groups are bonded to the α-carbon atom. On the other hand, it was difficult to distinguish m-C5 and v-C5 from i-C5 by the present method, but since such cases may be practically extremely rare, it is not necessary to distinguish them in the primary test for NBS.

There have been attempts to differentiate between i-C5 and p-C5 in NBS. Although the column and mobile phase are required to separate the isomers, LC–MS/MS analysis is useful as a second-tier test [25,27,28,29,30,31]. Shigematsu et al. reported a method for a second-tier test for the determination of isovalerylglycine using FIA-TMS. In this method, isovalerylglycine and pivaloylglycine were distinguished by the product ion, which was generated predominantly from quasi-molecular ions of isovalerylglycine butylester but apparently not from those of pivaloylglycine butylester [32,33]. Although the butanol-derivatization of DBS samples is needed in this method, it does not require column separation, and the time required for analysis is short. Recently, Maeda et al. reported a second-tier test method for excluding false positives derived from p-C5 using FIA-TMS. They distinguished p-C5 from other isomers of C5-acylcarnitine using acylcarnitine esterase to hydrolyze acylcarnitines [34]. This method has an advantage that the samples left over from the primary test can be used and no column separation is needed. In contrast, our method does not require a secondary test and has the advantage of being able to distinguish between i-C5 and p-C5 at the primary test. Our method can accurately distinguish i-C5 from p-C5 with only setting reference ion (*m*/*z* 246.2 > 187.1 CE: −15 V) in analytical methods and calculation of i-C5 and p-C5 scores in cases of elevated C5. In NBS, the samples above the cutoff value are retested for confirmation. Our method does not require any special equipment or pretreatment such as column separation. In this study, statistical analysis showed that the results of the initial test alone were sufficient to discriminate between i-C5 and p-C5, but a more accurate determination would be possible by averaging the results of the initial and confirmatory tests.

At present, it is difficult to distinguish leucine from isoleucine, C4-OH from C3-DC, C5-OH from C4-DC, and C5-DC from C6-OH by the non-derivatized FIA-TMS method. Although there are few practical problems in not being able to distinguish them at present, our method may reveal false-positives previously unknown by distinguishing isomers. Although we believe this method is promising for application in NBS programs, the limitation of this study is that the long-term stability of the reference ion ratios and changes in the false positive rate in multiple NBS laboratories were not validated. Further verification and accumulation of evidence by applying the method on a relatively large scale are expected in the future.

## Figures and Tables

**Figure 1 children-09-00694-f001:**
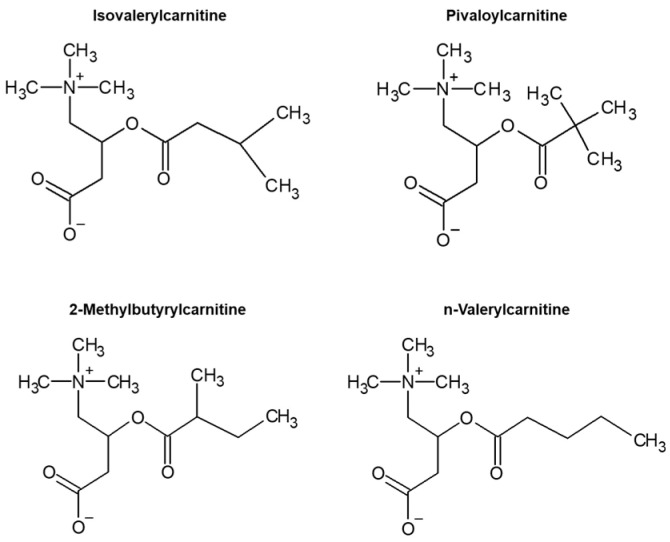
Structure of C5-acylcarnitine isomers.

**Figure 2 children-09-00694-f002:**
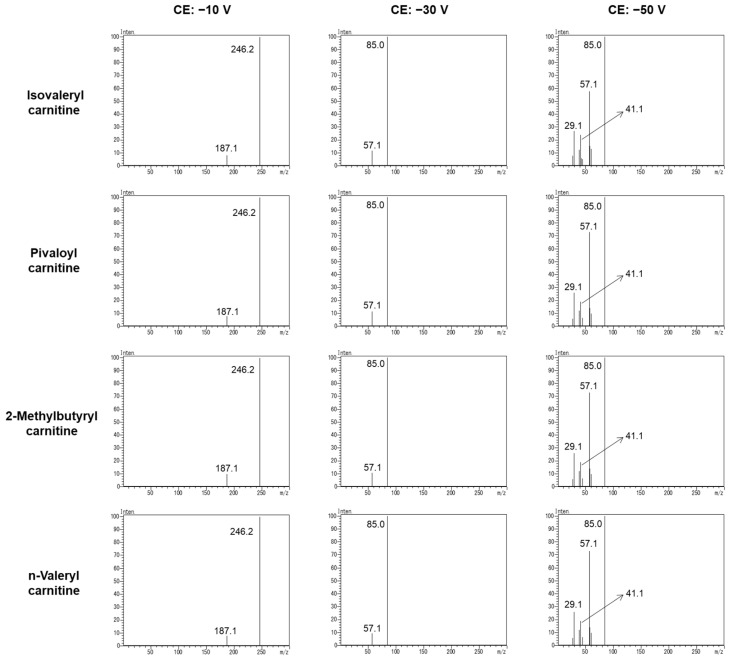
Product ion spectra of C5-acylcarnitine isomers.

**Figure 3 children-09-00694-f003:**
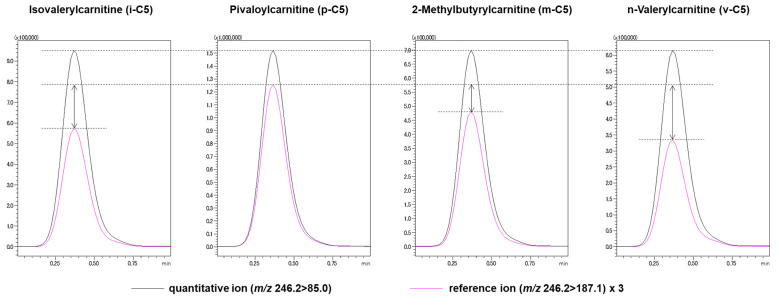
MRM chromatograms of C5-acylcarnitine isomers.

**Figure 4 children-09-00694-f004:**
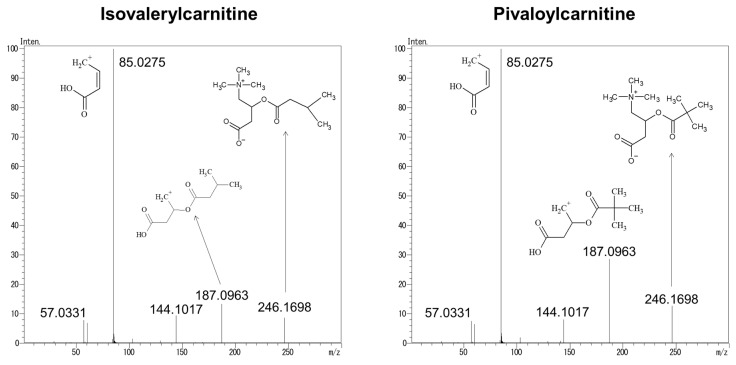
Product ion spectra of i-C5 and p-C5 using LC–QTOFMS.

**Table 1 children-09-00694-t001:** Analysis results of 15 DBS samples.

Case	% of Total C5-Acylcarnitine Isomers				C5 Conc. (μmol/L)	i-C5 Score	p-C5 Score
	i-C5	p-C5	m-C5	v-C5			
Samples from cases who had pivalate-containing antibiotics							
1	12.2 ± 1.1	79.7 ± 1.4	8.1 ± 0.7	0	0.66 ± 0.01	62.5 ± 5.5	97.1 ± 2.3
2	9.6 ± 0.5	90.4 ± 2.7	0	0	0.96 ± 0.02	37.1 ± 5.7	82.0 ± 4.1
3	0	100	0	0	1.31 ± 0.01	57.9 ± 6.7	96.5 ± 4.4
4	0	100	0	0	1.33 ± 0.01	58.8 ± 2.1	97.7 ± 1.5
5	0	100	0	0	1.37 ± 0.01	50.0 ± 2.4	91.3 ± 1.8
6	0	100	0	0	1.64 ± 0.01	55.6 ± 0.5	95.4 ± 0.4
7	0	100	0	0	1.67 ± 0.01	55.3 ± 1.8	95.2 ± 1.3
8	0	100	0	0	1.91 ± 0.02	55.6 ± 3.3	95.4 ± 2.4
9	0	100	0	0	3.73 ± 0.03	58.5 ± 1.3	97.5 ± 0.9
10	0	100	0	0	3.84 ± 0.03	41.7 ± 1.5	85.3 ± 1.1
11	0	100	0	0	10.98 ± 0.08	60.9 ± 0.9	99.2 ± 0.5
Samples from cases with isovaleric acidemia							
12	100	0	0	0	2.57 ± 0.02	91.1 ± 1.9	78.9 ± 1.4
13	100	0	0	0	2.67 ± 0.03	93.6 ± 4.7	77.1 ± 3.4
14	100	0	0	0	5.31 ± 0.04	97.3 ± 1.7	73.4 ± 2.3
15	99.9 ± 7.0	0.1 ± 0.0	0	0	13.72 ± 0.10	99.0 ± 0.5	72.9 ± 0.8

## Data Availability

Data are available from the corresponding author on request.

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
