# Peer review of "A Simple Flow Injection Analysis–Tandem Mass Spectrometry Method to Reduce False Positives of C5-Acylcarnitines Due to Pivaloylcarnitine Using Reference Ionsâ€"

_children, 2022, doi:10.3390/children9050694_

Round 1
Reviewer 1 Report
Dear Authors,
I find this an interesting manuscript, in which you have presented your research clearly. I have a few suggestions, which (in my opinion) will make the manuscript more complete:
- Line 63-65: This is a general statement, which is not related to IVA specifically. Can the authors include some statistics from their (or another newborn screening programme) regarding the number of false positive referrals for IVA because of use of antibiotics?
- Line 95: is a 3 mm punch used or 3.2 mm punch?
- Lines 167-170: reference should be added to support these statements.
- Line 267: could you explain a little more about the practical implications of these 'minor changes'? What would a programme using NeoBase need to do in order to implement your proposed changes in their programme. Why has this not be addressed by the manufacturer of the kit?
- Lines 273-275: Could you explain in a little more detail the impact this (inability to separate isomers using FIA TMS) has on programmes, giving specific examples?
- English could be improved, for example:
- Line22: Remove 'Then"
- Line 26: Add 'elevated' before C5
- lines 56-58: '....SBCADD is not included in NBS panels because it is an asymtomatic condition with unknown clinical consequences in the longer term...'
- line 72: '....because it required additional analysis' change to 'to avoid additional LC-MS/MS analysis'
Reviewer 2 Report
The authors present a methodology that allows the differentiation between i-C5 and p-C5 in FIA-LC-MS as used in newborn screening. This may be very usefull in newborn screening as it makes it possible to detect false positive IVA results that are caused by p-C5. This is highly relevant because the current methods used for differentiation generally call for a 2. tier method using chromatography, making it a slow and resource intense approach. The differentiation is based on the difference in ion ratios for signature fragments. The method is really only able to distinguish between p-C5 and the other three isobaric compounds. However, this solves a major issue in newborn screening (i.e. false positives due to the use of pivalate compounds).
The manuscript is well written and provides a method that may readably be tested in newborn screenings laboratories. There are, however, two issues that the authors must address before it may be considered ready for publication.
- In section 3.4 second paragraph (starting on line 214), the authors calculate the number if analyses required to measure the i-C5 and p-C5 score with an accuracy of ± 5%. I find this section hard to follow. Does this refer to the number of injections per sample or the number of scans that are necessary to accurately determine the scores? How is this to be understood in practical terms?
- Although the method appears promising, it is important to consider the fact that the prevalence of IVA is approximately 1 in 250 000 live births. Many labs run for years between true positive IVA cases. Thus, the experimental set-up described in the manuscript does not reflect the situation in a typical screening lab. The experimental set-up does not investigate how stable the ion ratios are over time and how well this works on borderline false positive IVA cases (i.e. C5 values close to the cut-off). Also, they have not investigates how other acyl-carnitine related inborn errors of metabolism may affect the resutls. These limitations should be mentioned in the discussion. Basically, the study should be presented as a proof of concept, not as a method ready for implementation.
Round 2
Reviewer 1 Report
I have no further comments, other that to congratulate the authors on publication of their work.